# Integrated SDN-NFV 5G Network Performance and Management-Complexity Evaluation

Nico Surantha [1,2,*] and Noffal A. Putra [1]

1   BINUS Graduate Program—Master of Computer Science, Computer Science Department, Bina Nusantara University, Jakarta 11480, Indonesia
2   Department of Electrical, Electronic and Communication Engineering, Faculty of Engineering, Tokyo City University, Setagaya-ku, Tokyo 158-8557, Japan
*   Correspondence: nico.surantha@binus.ac.id

**Abstract:** Digitalization is one of the factors that affects the acceleration of the application of telecommunications technologies such as 5G. The 5G technology that has been developed today does not yet meet different performance and manageability standards, particularly for data center networks as a supportive technology. Software-defined networking (SDN) and network function virtualization (NFV) are two complementary technologies that are currently used by almost all data centers in the telecommunications industry to rectify performance and manageability issues. In this study, we deliver an integrated SDN-NFV architecture to simplify network management activities in telecommunication companies. To improve network performance at the computing level, we performed a modification of a networking system at the computing level, underlying NFV devices by replacing the default virtual switch with a data plane development kit (DPDK) and single root I/O virtualization (SR-IOV). This study evaluated the proposed architecture design in terms of network performance and manageability. Based on 30 days of observation in prime time, the proposed solution increased throughput up to 200 Mbps for the server leaf and 1.6 Gbps for the border leaf compared to the legacy architecture. Meanwhile, the latency decreased to 12 ms for the server leaf and 17 ms for the border leaf. For manageability, we tested three different scenarios and achieved savings of 13 min for Scenario 1, 22 min for Scenario 2 and 9 min for Scenario 3.

**Keywords:** Software-Defined Network; Network Function Virtualization; network architecture; telecommunication; 5G

## 1. Introduction

5G technology is one of the main drivers supporting the needs of various vertical industries such as automotive, manufacturing, energy, healthcare, media, and entertainment. 5G technology developed today that uses a "one-size-fits-all" architectural approach cannot meet the different performance standards to be applied in each industry in terms of scalability, latency, reliability, and availability [1]. To efficiently facilitate vertical-specific cases as the demand for legacy services increases over the same network infrastructure, it is acceptable that 5G systems will necessarily involve architectural upgrades relative to existing deployments [2].

When providing new services, for instance in the case of smart cities and Internet of Things (IoT), the 5G architecture should enable agility, speed, and cost-efficiency. 5G networks must also provide multi-service, multi-tenancy, and multidomain support. The main concepts for achieving an architecture with speed, agility, and cost efficiency are virtualization, softwarization, and programmability [3]. These three concepts are included in the new paradigm of computer networks: network software and network slicing [4].

Network softwarization (Netsoft) (San Jose, CA, USA; Palo Alto, CA, USA) is a paradigm for increasing network automation by enhancing network programmability. Netsoft refers to the evolution of the network industry to design, deploy, and maintain

network devices/network elements via software programming [5,6]. This allows flexibility to redesign network services in order to optimize costs and enable self-management capabilities to manage the network infrastructure [7].

Network softwarization can provide the modularity needed to create multiple (virtual) logical networks through software defined network (SDN) and network function virtualization (NFV) technology. This logical network is also known as network-slicing [8]. Network slicing is an end-to-end (E2E) logical network running on top of a physical or virtual network, isolated from each other, with independent management and control that can be created on demand [9].

The problems that occur in the application of the main supporting technology for the 5G networks in telecommunications companies are speed and agility limitations. The number of hops (jumps) per network element in the data center becomes a bottleneck that causes high latency and reduced throughput resulting from the network in the data center, particularly for traffic that leads to virtualized services [10]. In addition, another problem is agility. When we are going to create a new application or service that will be stored in the data center, either as bare metal or virtualized, we will go through a fairly long process until the application or service can be accessed [11].

We showed that SDN and NFV can be simplified by integrating with both controllers [3]. The two controllers in question are the SDN and NFV, which can be integrated with the API that occurs in both backend controllers. When they integrate, activities that are usually carried out on one controller can be carried out on the other. In addition, to enhance the performance of SDN and NFV, we can also simplify traffic at the computational level by reducing hop traffic which is currently at the computing level; almost all use virtualization. In the implementation, we used DPDK and SR-IOV to improve the performance at the computational level. DPDK uses the kernel to process packets, which puts pressure on the kernel to process packets faster because the speed of the network interface card (NIC) increases [12]. DPDK is also a better choice than other methods such as vanilla PF_RING and 6WINDGate in terms of open source, security and bypass network stack [13].

There are several techniques for bypassing kernels in the network stack for efficient packet delivery [14]. Kernel bypassing techniques are divided into 3 categories: User-space Packet I/O, User-space TCP/IP stack and Hybrid network stack. From these three categories, user-space packet I/O removes the cost of packet transfer between the NIC and the host stack. This is an advantage for applications that run on it because it no longer requires high-level protocols processing (e.g., switch, routers, firewalls, etc.). Netmap, DPDK, mTCP, and IX are a type of User-space Packet I/O. DPDK is chosen because it uses more radical optimization techniques to pursue extreme packet I/O speed which is needed to improve packet processing performance [15]. Modifications are then made in the network part of computation, which is interfaces (PCI Express). This is also done as a way to improve the performance of the computing system to support the services running on it. SR-IOV was chosen because it specifies hardware enhancements that reduce the interactions between a hypervisor and a virtual machine (VM) in order to optimize the VM's data processing performance [16,17]. SR-IOV uses a technique to circumvent the hypervisor and has direct access to the NIC's virtual network function (VNF), allowing for increased throughput. Therefore, we implemented all these architectures in the data center network environment and evaluated their performances and manageability. We used the Cisco application centric infrastructure (ACI) as the SDN infrastructure and the VMware TCI as the NFV infrastructure. Cisco ACI is an SDN data center infrastructure that is commonly used for creating a data center fabric [18]. Latency and throughput parameters are utilized as indicators of the network performance [19]. These parameters indicate the speed and reliability of the communication process while serving customers. Therefore, the main goal of this study was to prove that the application of DPDK and SR-IOV can improve the performance of SDN and NFV infrastructure integration by conducting traffic simulations and testing, and to prove that integration between SDN and NFV can simplify daily operational activities. We have succeeded in proving that integration between SDN

and NFV is a convenient architecture for supporting 5G technology from simulation results, as will be clarified later.

In this study, we focus on the integration between SDN and NFV to support 5G technology. This integration enables the acceleration of problem-solving time in various operational scenarios. In addition, we obtain network performance for data center network traffic. The main contributions of this study are summarized as follows.

(1) An integrated SDN-NFV architecture to simplify network management activities.
(2) Modification of the networking system at the computing level underlying NFV devices by replacing the default virtual switch with DPDK and SR-IOV.
(3) The evaluation of proposed architecture compared to legacy architecture (SDN-only) in terms of network performance and manageability.

The remainder of this paper is organized as follows. Section 2 explains the related work for this research and some related theories for strengthening our understanding before entering the next section. Section 3 clarifies the legacy architecture using the proposed architecture. Section 4 quantitatively compares the different network performances and manageability after integration between SDN and NFV. This proves that integration between SDN and NFV is the best choice for data center networks in with compliance with 5G technology. Finally, Section 5 concludes the paper and highlights future work.

## 2. Related Work

The underlay and overlay network concept, software-defined network and network function virtualization concept, integration of SDN and NFV in 5G technology, and independent SDN and NFV in 5G technology were included in this study. Hence, the related work is outlined.

### 2.1. Underlay and Overlay Network Concept

Spine-leaf architecture as shown in Figure 1 has the ability to answer the needs of modern data centers. It provides low latency and minimum packet loss by enabling the addition and deletion of network nodes on demand. This also has an impact on the evolution of the data center (DC) architectural design and network infrastructure, which also plays an important role in that change, as mentioned in [9]. Several aspects of this new paradigm need to be considered:

(1) Flexibility: Allows mobility from anywhere.
(2) Resilience: Maintain service even in damaged conditions.
(3) Multitenant Capacity: Better segmentation of workloads in the network.

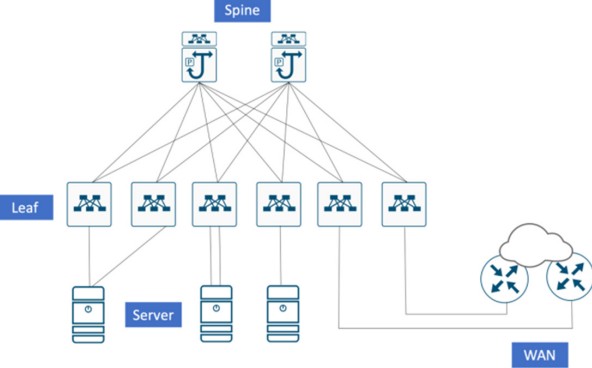

**Figure 1.** Spine-leaf architecture.

One of the main problems when implementing modern network infrastructure, both physical and virtual, is the limited number of IP addresses that leads to the search and reuse of available IP addresses when they are not in use, or on network devices that are no longer in use. Another limitation of traditional networks is the limitation of virtual

LANs (VLANs) which can only be capable of covering 4094 VLANs. This makes it difficult to implement networks on a large scale, such as telecommunications service providers and data center networks. When using virtual extensible VLAN (VXLAN) technology, we can address this shortcoming because VXLAN can cover approximately 16 million IDs. A VXLAN can also optimize logical addressing because it connects different sites with the same IP addressing block. A multitenant DC allows the allocation of service provision, and is isolated to several different users by providing a specific segment for each customer [20].

The use of multitenant DCs has expanded as a result of the virtualization platform. They are now found not only in large service providers, but also as part of network infrastructure. One way to achieve this is to implement an overlay network architecture that encapsulates packets. The idea is to implement a network "on top" of others that have been created by tunneling between links through an established network infrastructure, commonly called an underlay network.

### 2.2. Software-Defined Concept

The high cost of procuring this switch is associated with the hardware and software development costs in conventional technology and is a trigger for the development of data center networks. An important component of operational expenditure costs is the per-device software license and maintenance [21]. In software-defined networks, the majority of software control planes are removed from the route processor, and instead reimplemented for execution on external commodity hosts that are under the control of the internet and cloud service providers [22]. The conclusion is that the cost of developing and maintaining SDN software will be lower than that of maintaining traditional switches and models designed for SDN networks [23].

In traditional networks, control plane protocols (such as routing protocols) and forwarding table computational algorithms are executed on the route processor in the chassis. For example, when we use open shortest path first (OSPF) routing, the router exchanges messages with neighboring routers in the same domain, studies the topology, and runs the Dijkstra algorithm to calculate the shortest route and complete the routing table. The routing table is then copied from the route processor to the forwarding table stored in the OSPF database [24].

Figure 2 shows an SDN solution that reduces the routing cost by removing the control plane software from the route processor and replacing it with a simple control agent, such as OpenFlow, running on the switch processor. As depicted in Figure 2, the control plane algorithm for actions such as forwarding table computation is executed on the SDN controller installed on the server. The calculated forwarding table is then retrieved to the switch via a protocol, such as OpenFlow or NETCONF.

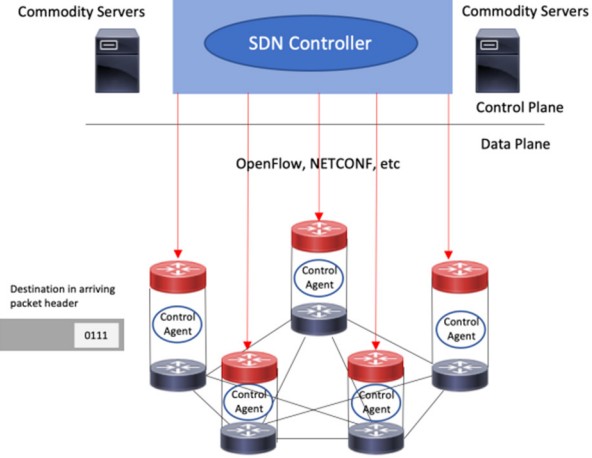

**Figure 2.** SDN solution.

### 2.3. Network Function Virtualization Concept

The basic functions of the data plane, such as searching and forwarding packet headers, can be implemented in software on commodity servers, for example, commodity servers. Moreover, new servers that apply the latest processor, disk, and memory technology and are more energy efficient can replace older servers, where network functions (NF) are usually run. In addition, NF can be extended to run on several different operating systems (OS), virtual machines (VM) hypervisors and containers. NFV-based network switches and middleboxes are anticipated to incur less CAPEX and OPEX due to industry competition in the commodity server market [25].

Figure 3 illustrates that by using a VM, several network functions such as firewalls, routers, Deep Packet Inspection (DPI) and monitoring tools can be run on a single server [20]. Compared to traditional solutions, where each network function is implemented on its own equipment, sharing the same physical server for multiple network functions increases CPU utilization and energy efficiency. In a relatively small campus network, it is possible to implement all required network functions on a small number of physical servers containing multiple VMs.

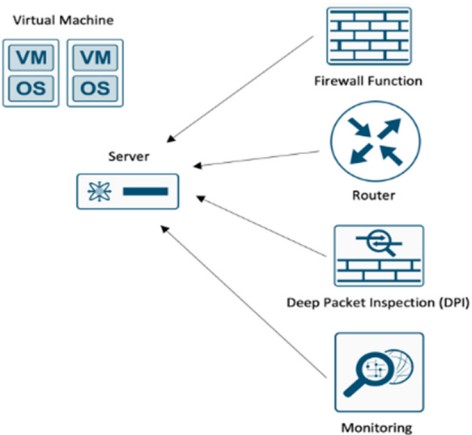

**Figure 3.** Network function virtualization.

### 2.4. Integration SDN and NFV in 5G Technology

In a written paper [26], Khan et. al. tried to solve the complex challenge of integrating SDN and NFV by proposing software-defined network virtualization (SDNV) architectural framework. The SDNV framework combines the SDN concept of splitting data and control planes with the NFV principle of decoupling service functions from infrastructures, providing a holistic view of SDN and NFV integration. To illustrate the relationship between SDN and NFV principles, Khan et al. first discuss how SDN and NFV might benefit the model. Then, they proposed the SDNV framework architecture, which provides a high-level picture of SDN and NFV integration. Combining SDN and NFV would have many benefits, but integrating them is a challenge. There are two possible architectures for such integration: either the controller or the switch interacts with virtualized network functions (VNFs). Fahmin et al. refer to the former as NFV under the controller (NFV-C) and the latter as NFV alongside the controller (NFV-AC). To the best of our knowledge, no analytical model exists for mathematically examining the performance of such architectures [27].

In other articles, there is some research to compare the performance of software switches which relate to the integration of SDN-NFV. Still, adding a high-speed framework like DPDK or SR-IOV will accelerate the forwarding speed of the switch. Switches are important because most of the network components used in SDN and NFV platforms are routing capable switches. Zhang et al. investigate their respective design spaces and evaluate their performance in four representative test scenarios. Each scenario corresponds to a unique instance of NFV traffic routing between NICs and/or VNFs. Their experimental findings demonstrate that no single software switch is optimal in all circumstances. There-

fore, it is essential to select the one that is most suited to a given use case. The presented results and analysis improve comprehension of design tradeoffs and identify potential performance bottlenecks in software switches [28].

From related work articles on integration that have been described, the most significant difference is in the article that the authors made describing the results of the integration seen from the time of operational convenience. In addition, the comparison made is a comparison between SDN only and SDN-NFV. Meanwhile, other articles on SDN and NFV integration are more focused on the integration process and performance.

In Table 1, the comparison summary of SDN-NFV integration articles which are related to this experiment is presented to give a better understanding about this topic.

**Table 1.** Comparison of SDN-NFV integration which related to this experiment.

| Title | Architecture | Contribution | Evaluated Parameter |
|---|---|---|---|
| Software-Defined Network Virtualization: An architectural framework for integrating SDN and NFV for service provisioning in future networks [26] | Software-Defined Network (SDN), Network Function Virtualization (NFV) and Software-Define Network Virtualization (SDNV) as proposed solution | Duan et al. tackle the challenging problem of integrating SDN and NFV in future networks by presenting an architectural framework that combines the key principles of both paradigms. Performance analysis and modeling of SDN and NFV integrating on the performance of the interaction effect of SDN and NFV controller (under or aside controller). | Framework & Architecture |
| Performance modeling of SDN and NFV under or aside the controller [27] | Software-Defined Network (SDN), Network Function Virtualization (NFV) | | Packet delay |
| Comparing the performance of state-of the art software switches for NFV [28] | Software-Defined Network (SDN), Network Function Virtualization (NFV), Data plane development kit (DPDK) and single root I/O virtualization (SR-IOV) | Bring a better understanding of design tradeoffs and identify potential bottlenecks that limit the performance of software switches. | Throughput and Latency |
| Integrated SDN-NFV 5G Network Performance and Management-Complexity Evaluation (Proposed) | Software-Defined Network (SDN), Network Function Virtualization (NFV), Data plane development kit (DPDK) dan Single root I/O virtualization (SR-IOV) | • An integrated SDN-NFV architecture to simplify network management activities. <br> • Modification of the networking system at the computing level underlying NFV devices and the evaluation of proposed architecture compared to legacy architecture (SDN-only). | Daily operational task time, Throughput & Latency |

### 2.5. Independence of SDN and NFV in 5G Technology

SDN and NFV are closely related and can be combined efficiently to support 5G technology. NFV is more complementary to SDN, and is not interdependent. Network functions are virtualized and distributed without SDN [29]. Meanwhile, functions that are not virtualized are controlled by SDN. The main properties of SDN and NFV apply when NFV modifies hardware network elements using software that resides on the server. SDN functions are performed separately by exchanging the network functions with generic appliances and servers. In addition, NFV will make adjusts to the use of servers and switches in SDN.

Therefore, it can be concluded that the solution in this journal is that SDN and NFV work independently, even though SDN is convenient to use and operate with NFV. SDN manages the network complexity in software that is run centrally (coordination, access, and programmability) [30]. SDN can also act as an orchestrator to configure the devices it manages from the server controller through a software mechanism. NFV provides more configuration options for the software functions above.

## 3. The Proposed Architecture

This section may be divided by subheadings. It should provide a concise and precise description of the experimental results, their interpretation, as well as the experimental conclusions that can be drawn.

### 3.1. Requirement Gathering

Using the Plan Design Implement Operate Optimize (PDIOO) framework [31], the authors will design a best practice networks data center in telecommunications companies to support 5G implementation. This includes data traffic for inbound-outbound virtualization-based services.

For the proposed solution, the authors use a top-down approach to target staff at a higher level, such as a general manager, to understand what the business goals and constraints are, and a manager to understand the technical goals of implementing SDN and NFV technology in this company, as the mapping presents in Table 2.

**Table 2.** Technical Goals to Business Goals Mapping.

| Business Goals | Technical Goals |
|---|---|
| • Increase the user-experience of telecommunication company customers | • Decreased latency on intra-service or inter-service data traffic based on virtualization.<br>• Improved throughput performance for virtualization-based service application data traffic. |
| • Make it easier for companies to manage data center infrastructure. | • Ease of day-to-day operations in managerial (check, change and troubleshoot). |

### 3.2. Design

In this study, the initial condition of the architecture in the data center was using SDN, but this SDN was still independent, so that the computing level below it was still running independently, as shown in Figure 4. The SDN used was the Cisco Application Centric Infrastructure (ACI). The computing level consists of the servers used to serve users.

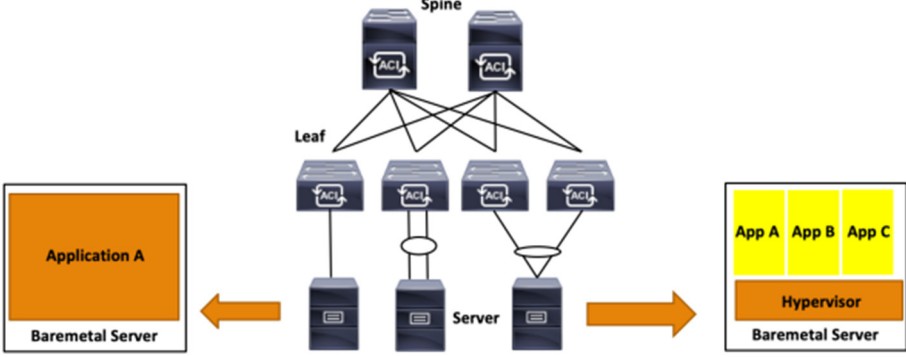

**Figure 4.** Independent SDN and computing system.

On the server, many applications run either directly on the server or over the virtualization hypervisor. Hypervisor virtualization using VMware ESXi (Cisco: San Jose, CA, USA; VMware: Palo Alto, CA, USA). The applications are interconnected using the provided vSwitch standard [32], as shown in Figure 5.

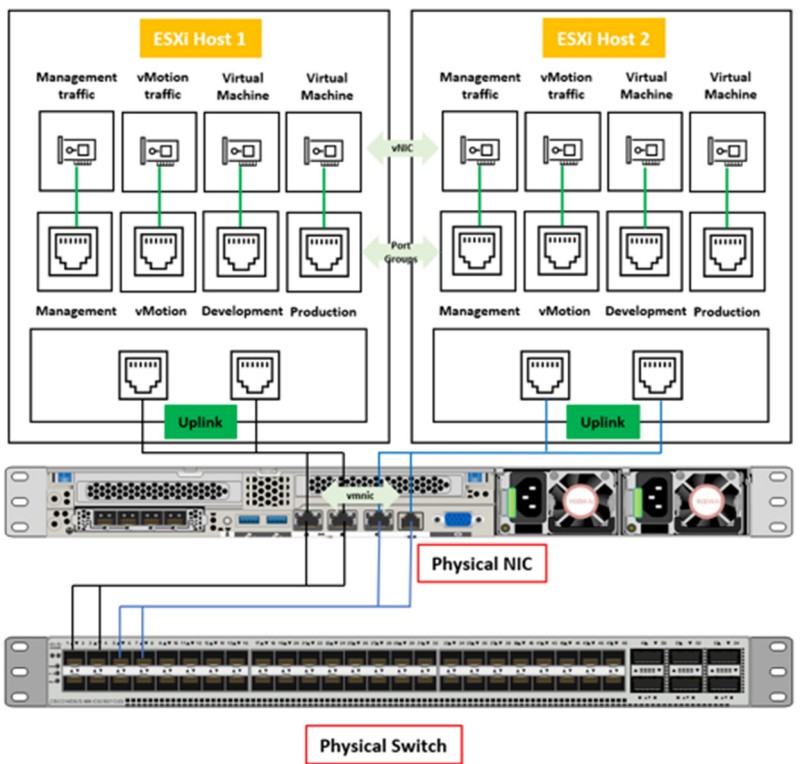

**Figure 5.** vSphere standard switch architecture [25].

The following is a logical design that will be applied to telecommunications companies based on needs analysis and technical goals, as well as a literature review by the authors. The two switches above are spine and leaf SDN, which are commonly referred to in the unit, namely Fabric as shown in Figure 6. The endpoint devices attached to the leaf switch are servers with various functions, some of which function as SDN controllers, virtualization servers, network function virtualization computing servers, and external routers for external connections.

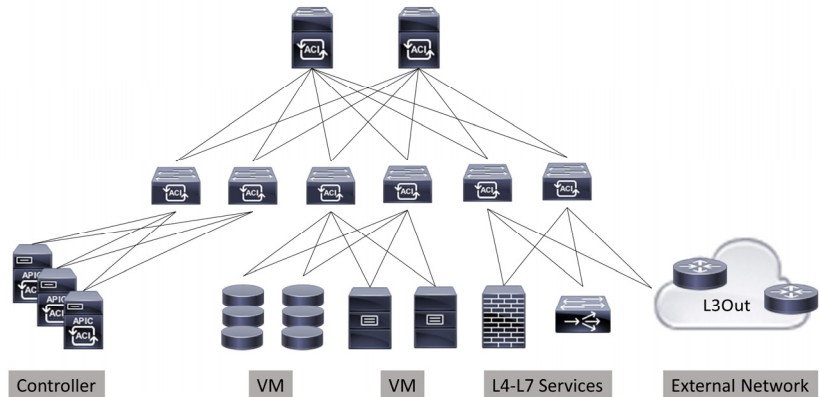

**Figure 6.** Logical design of SDN and NFV.

Next, we provide a further explanation of the SDN topology that has been integrated with NFV. In this design, the SDN Controller (Cisco APIC) is integrated with the VMware

TCI's VMware vCenter, which contains the Controller and management service of a centralized virtualization platform, as shown in Figure 7.

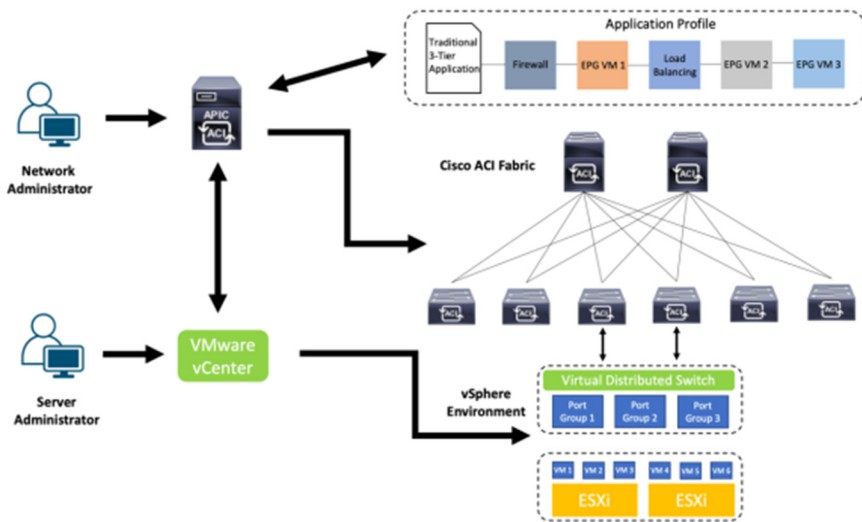

**Figure 7.** The proposed integration of SDN controller with NFV controller and management service.

In addition, the implementation of DPDK and SR-IOV is carried out at the compute level, where each virtual machine is determined using one of these techniques to forward data packets. For virtual machines that have many north-south connections, SR-IOV is applied to the computing server because it uses hardware-based switching techniques and is not limited by Open vSwitch (OVS), which is a purely software-based solution [33]. For the east-west connection, DPDK is recommended because it does not utilize the physical NIC as depicted in Figure 8, where data traffic will not be forwarded until the physical NIC, which is the purpose of the data packet, is still on the same server (not different hardware) [12].

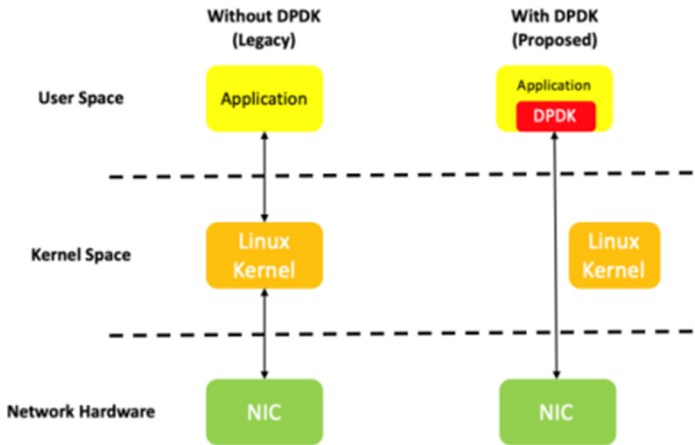

**Figure 8.** DPDK architecture.

The data plane development kit (DPDK) uses the kernel to process packets, which puts pressure on the kernel to process packets faster because the speed of the network interface card (NIC) increases [12]. We use the bypassing kernels method for efficient packet delivery. This feature involves processing packets in the user space instead of the kernel space; DPDK is one such technology. SR-IOV uses a technique to circumvent the hypervisor and has direct access to the NIC's virtual network function (VNF), allowing for increased throughput, as shown in Figure 9 [33].

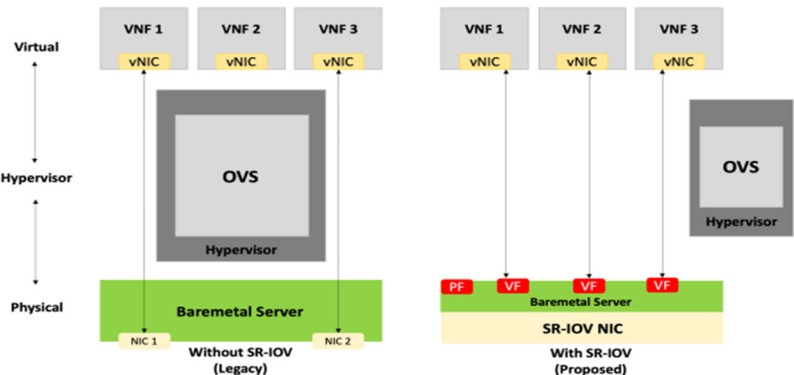

**Figure 9.** SR-IOV architecture.

The authors use a top-down approach to target staff at a higher level, such as a general manager, to understand the business goals and constraints of a manager to understand what the mapping presented in Table 3.

**Table 3.** Design Decision to Technical Goals Mapping.

| Technical Goals | Design Decision |
|---|---|
| • Decreased latency in virtualization-based intra-service or inter-service data traffic. | • Designing best practice SDN and NFV network architectures so that all SDN and NFV capabilities can be utilized to the fullest. |
| • Improved throughput performance for virtualization-based service application data traffic. | • Implement appropriate network protocols and the right SDN and NFV features so that the amount of throughput generated is as expected. |
| • Ease of daily operations in performing managerial (check and troubleshoot). | • Perform integration between SDN and NFV so that control over these technologies is in one dashboard. |

*3.3. SDN-NFV Integration*

The first step that must be taken to start the integration of SDN and NFV Controller is to initiate connectivity between controllers, as shown in Figure 10. When connectivity is connected, credentials can be exchanged via the SDN controller dashboard. Then after authentication is complete, we need to create a Virtual Distributed Switch as a Layer 3 (Routing) Virtual Network Function (VNF) interconnect that is managed by the controller.

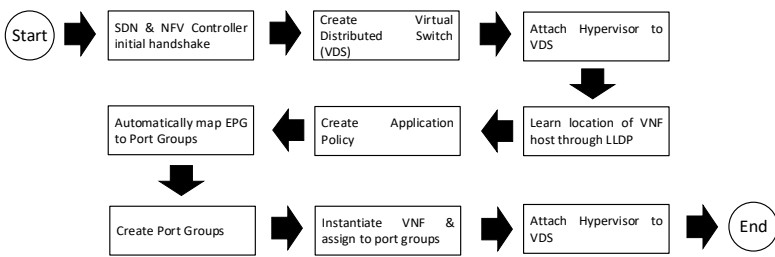

**Figure 10.** Integration process.

Since this VNF stands on the hypervisor, we also need to connect it to the previously created VDS. Parallel to this process, the SDN and the NFV controller exchange endpoint-related information and also automatically map endpoints to port groups. After that, we have to create an application policy as an access card from each endpoint to reach the destination. Next is to adjust the port groups that have been mapped, whether they are

appropriate or not. When appropriate, the created VNF instance should be assigned to the port groups.

## 4. Evaluation and Result

The objective of this section is to ensure that the integration of SDN and NFV is suitable for data center network infrastructure. This architecture also clarifies the benefits of using DPDK and SR-IOV at the computing level and the data center infrastructure for improving network performance. Therefore, we used throughput and latency parameters to demonstrate that this integration can enhance network performance. This integration and modification in the computing level is the one that possesses less latency and higher throughput, compared to other implemented SDN that are independent. In addition, we tested manageability by measuring the duration of changes that are usually made by personnel in carrying out daily operations.

### 4.1. Simulation Parameter

In this section, we describe the implementation of Cisco ACI as the SDN platform used for testing. This device is a production device that is used directly to serve customers. For the virtualization platform, we use VMware ESXi as the base hypervisor, which will later be used to run the virtualized network functions. At the compute and hypervisor levels, we modified the network on the server to improve the performance of the server when forwarding data. We replaced the standard switch at the hypervisor level with DPDK and SR-IOV technologies. In addition, we also integrated the SDN controller, in this case the Cisco application policy infrastructure controller (APIC) with VMware vCenter as the NFV controller. For the manageability test, we integrated the controllers by using the APIs provided by each platform. Tests were also carried out by each administrator in several previously made scenarios.

Table 4 clarifies the main simulation parameters used in each scenario to provide a better overview of each network for making the best decision. The number of SDN controllers is the same as 3 between Independent SDN and integrated SDN NFV, because there is no simulation related to the addition of SDN controllers. Then, there are 2 NFV controllers which only applies to integrated SDN NFV, because Independent SDN does not use NFV. The number of VM controllers is the same, because the calculation is based on the number of hypervisors on the monitored server. Meanwhile, the number of switches is the same between Independent SDN and NFV SDN, because no simulation related to the physical network environment has been carried out.

**Table 4.** Simulation parameter.

| Parameter | SDN Independent | SDN with NFV Integration |
| --- | --- | --- |
| Number of SDN Controller | 3 | 3 |
| Number of NFV Controller | 0 | 2 |
| VM Controller | 10 | 10 |
| Number of Spine Switch | 2 | 2 |
| Number of Leaf Switch | 4 | 4 |
| Number of Server which monitored | 2 | 2 |
| Monitoring Duration | 30 Days | 30 Days |

### 4.2. Performance Evaluation

We evaluated the performance of the proposed architecture that integrates SDN and NFV with Cisco ACI (N9K-C9508 as Spine and N9K-C93180YC-FX as Leaf) and VMware TCI (Inside UCS Hyperflex Server with 10G FO Interface, 512 GB of RAM, and 32 TB SSD Storage). The evaluations were performed as follows.

(1) Previously, the throughput capacity generated from traffic communication that led to applications running on virtual machines was deemed less than optimal, because it only utilized the default virtual switch from the logical device platform virtualization.

Therefore, there is an evaluation that will compare how the throughput capacity and latency between data centers that only use SDN, with data centers that combine SDN and NFV for communication from outside to VM, or vice versa.

(2)  For this performance evaluation scenario, we used Cacti [34]. The parameters that will be compared are the inbound and outbound throughput capacity of the interface, that provides communication on virtualization-based service applications and their latency. This test was carried out for 30 consecutive days (including weekends) so that the comparison of the data can be seen more clearly with the previous one, and was carried out at 18:00–21:00 when data traffic was heavy (peak hours).

(3)  In this test, the network function that will be used as a testing tool is a virtual switch (VMware NSX as part of the VMware TCI) that is installed on the server to redirect cache traffic to applications accessed by customers.

Figure 11 shows the test topology, which is used to measures the throughput and latency from the implemented solution, where the monitoring server is placed in the DMZ block. The server installed Cacti as a monitoring tool, and Cacti was reachable by the server leaf and border leaf switch. Two interfaces are used as points of monitoring: interfaces to servers that have installed network function virtualization, servers with virtualized computing inside, and interfaces that face the Internet. The servers-facing interface is on the server leaf switch and the internet-facing interface is on a border leaf switch.

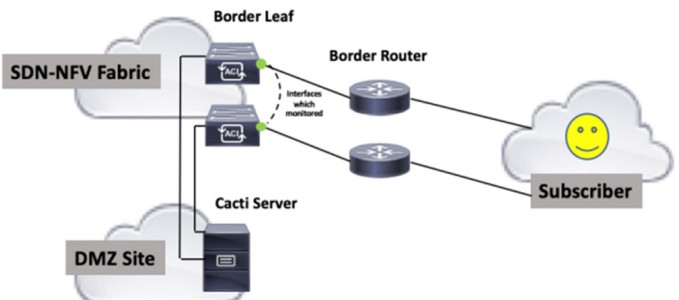

**Figure 11.** Test topology schematic using Cacti.

We compared the legacy architecture with the proposed architecture by calculating the average throughput and latency. We monitored the 10 selected interfaces on the leaf switch in the following manner:

(1)  3 interface in 2 server leaf switch which leads to monitored server.
(2)  2 interface in border leaf switch which leads to Internet or subscribers.

From the data that we collected for 30 days during the monitoring period before migration (legacy) and 30 days after implementation of the proposed architecture, we averaged per prime hour (18:00–21:00). Every hour, for example, at 18:00 we collected data from the interfaces that were selected for data collection which will be calculated later. The calculation is performed by calculating the average throughput at a certain hour: for example, we will calculate the average throughput and latency before migration at 18:00 on the leaf server, and we will calculate the average throughput and latency after migration at 19:00 on the border leaves.

Finally, we obtained the average of each section, for example, the server leaf at 18:00 or border leaf at 19:00 inbound or outbound to be compared to determine whether this integration and modification improves the performance of the data center network.

(a)  Throughput

The data displayed in the graphs in Figures 12 and 13 show that there is an increase in the throughput trend in both the input and output rates of traffic on the leaf switch server. The throughput increases values shown in the graphs in Figures 12 and 13 were 160 Mbps to 220 Mbps for the input rate and 170–230 Mbps for the output rate. The increase was due to a modification of the network computing system, namely the DPDK and SR-IOV on servers

filled with network function virtualization; thus, it is evident that the traffic cycle process between hosts on one server, as well as between hosts between servers (east-west traffic) becomes simpler and optimal according to the direction of data traffic communication. This leaf switch server also serves to communicate data traffic to the outside (north-south traffic) to be forwarded to the border leaf switch, but with an amount that is not more than the data traffic between servers.

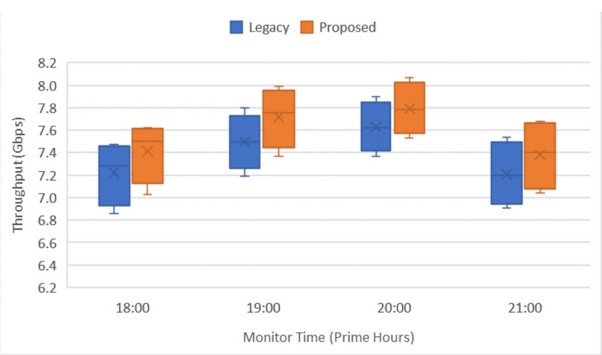

**Figure 12.** Server leaf switch throughput—All week comparison (input rate).

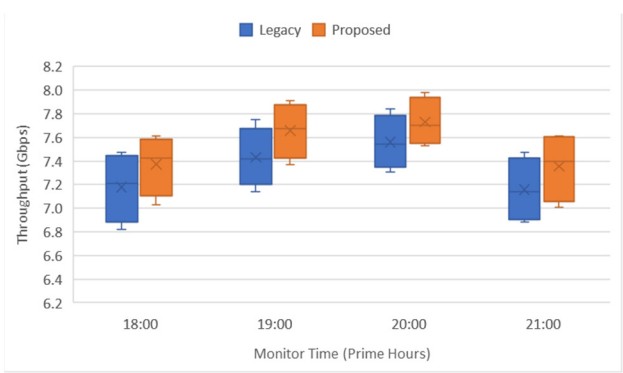

**Figure 13.** Server leaf switch throughput—All week comparison (output rate).

The data from the graphs in Figures 14 and 15 show the trend of increasing throughput values that occurred after the implementation of the proposed solution. The impact of the modification of the network computing system is not only visible at the server leaf switch level, but also at the border leaf switch. The throughput increase value shown in the graphs in Figures 14 and 15 is 1.22 Gbps to 1.8 Gbps for the input rate, and 670 Mbps to 1.1 Gbps for the output rate. Traffic data on the border leaf switch is an aggregation of all server leaf switches in the fabric, which can also be called the estuary of the server leaf switch. This border leaf switch is connected directly to the external network through an external router that has a direct connection with it. Therefore, data traffic on the border leaf switch is traffic to the outside network, both inbound and outbound. When viewed from the data displayed on the graphs in Figures 13 and 15, inbound traffic is more dominant in the border leaf switch data traffic because of the large number of subscribers (users) who make requests to servers in the fabric rather than requests from servers to the outside network or the Internet.

Based on Table 5, there is an increase in throughput on both the server leaf switch and border leaf switch, after the application of the new SDN-NFV architecture and modification of the network computing system on the server containing network function virtualization. In this SDN and NFV network architecture, we can see that the incoming and outgoing throughput began to creep up from 18.00 h and then ramped up again at 21.00 h at the site used for testing. If we compare the SDN-NFV network and SDN only, on average, we can clearly see that the SDN network has the ability to issue a smaller throughput compared to the SDN-NFV combination.

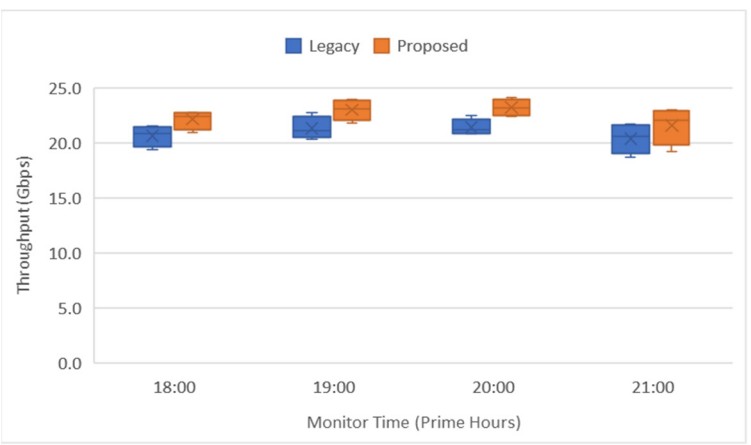

**Figure 14.** Border leaf switch throughput—All week comparison (input rate).

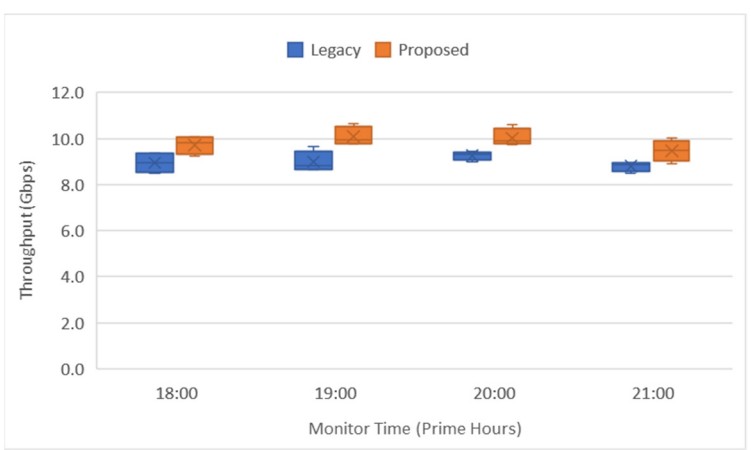

**Figure 15.** Border leaf switch throughput—All week comparison (output rate).

**Table 5.** Legacy vs. proposed throughput comparison table.

| Parameter | Mean Legacy | Mean Proposed | Deviation Standard Legacy | Deviation Standard Proposed |
|---|---|---|---|---|
| Server Leaf Input Rate | 7.39 Gbps | 7.58 Gbps | 0.206 | 0.208 |
| Server Leaf Output Rate | 7.33 Gbps | 7.53 Gbps | 0.197 | 0.193 |
| Border Leaf Input Rate | 20.97 Gbps | 22.51 Gbps | 0.502 | 0.750 |
| Border Leaf Output Rate | 9 Gbps | 9.83 Gbps | 0.194 | 0.285 |

This means that the SDN-NFV combination can serve more data requests from subscribers than the SDN alone. The greater the throughput value, the better because it can be faster and fulfills more subscriber traffic. In the research conducted by Nisar et al. [35], the researcher performed a simulation using an independent SDN, namely the Ryu controller and the OpenFlow switch. From this research, the peak throughput is not too large, which is around 25 Mbps. This is because the test uses a simulator with a workload that is not too heavy, and the endpoint is not a computing system with a certain load so that we cannot know the actual results. In our research, we conducted tests using real world SDN and NFV environments where the two technologies are integrated with each other down to the computing level. In addition, the load given to the endpoint system is also very large, so that we can see the actual peak performance of the combination of SDN and NFV. As an endpoint, we also use a real computing system so that it truly describes the actual conditions in the field.

(b)    Latency

From the data presented in Figure 8, it can be seen that there is a striking difference between the latency of the legacy and the proposed traffic data. The value of decreasing data traffic latency on the leaf switch server is 12.05 ms to 12.15 ms, as shown in Figure 16. This decrease in data traffic latency value is closely related to the optimization of hops through which data traffic passes, for example, if previously there was traffic data from host-1 to host-2 which turned out to be on one server, the traffic data would be forwarded out of the server to the leaf switch server. This is then reflected to the spine switch, and then returned to the leaf switch server until it returned to the server and host-2. Of course, the flow of data traffic such as this can be simplified by bypassing several data-forwarding mechanisms in the computing system, which in this study was carried out using the DPDK and SR-IOV on the server. With the application of DPDK and SR-IOV, the flow of traffic data between hosts and between servers can occur more simply and optimally to reduce the number of hops when data traffic passes. Optimization is performed by mapping the data traffic (between hosts/servers) with the network computing system that will be applied to the virtualized host/network function. Thus, the flow of data traffic and the networking computing system used is right on the target.

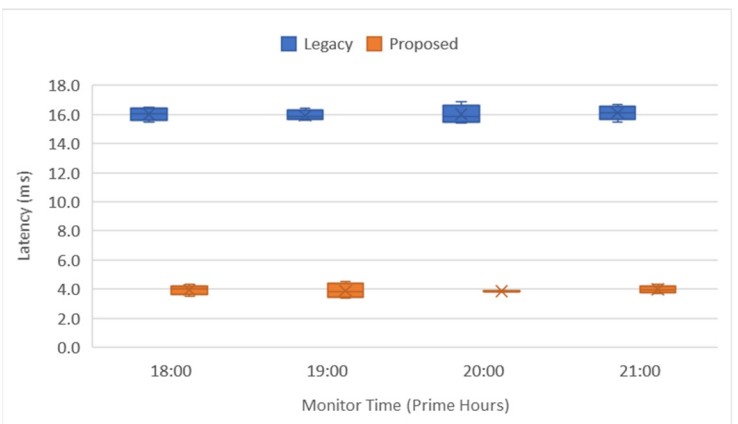

**Figure 16.** Server leaf switch latency—All week comparison.

The border leaf switch is the same as the explanation in the throughput section where the border leaf switch is the estuary of the server leaf switch, and the border leaf switch manages the outgoing traffic data or north-south traffic. The value of decreasing data traffic latency on the leaf switch server is 16.75 ms to 17.85 ms, as shown in Figure 17. A fairly large impact is also seen in the border leaf switch from the application of the new SDN-NFV architecture and modification of the network computing system on this server, because the number of hops passed by data traffic originating from subscribers (users) will also be reduced as well.

From Table 6, we can see the trend of increasing latency values after the implementation of the proposed solution. The latency value of SDN-NFV testing on the border and the leaf switch is influenced by the number of hops and the direction of the data traffic. The latency value on the leaf switch server is smaller than that of the border leaf switch, because data communication on the leaf switch server is limited from host to host on one server, host to host between servers, server to leaf switch, and server to border leaf switch, while the value latency on the border leaf switch has more hops because it also counts the hops to the subscriber in terms of the hops at the very end. However, the focus of this research is the comparison of the latency value of the SDN-NFV combination with SDN alone. On average, we can clearly see that the SDN network has a higher latency value than the SDN-NFV combination. This is because of the use of virtualization technology that can boost server performance in processing data. This means that the SDN-NFV combination can serve data requests from subscribers faster than an SDN alone. The smaller the latency

value, the better because it can be faster and can fulfill more subscriber traffic. Of course, a difference in performance is felt on the subscriber (user) side. This is in line with the business goals that the company wants to improve the user experience, so that users feel comfortable using company data services.

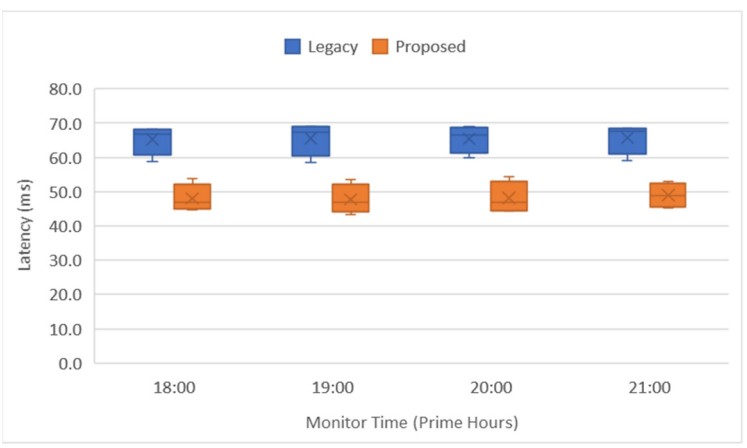

**Figure 17.** Border leaf switch latency—All week comparison.

**Table 6.** Legacy vs. proposed latency comparison table.

| | Mean Legacy | Mean Proposed | Deviation Standard Legacy | Deviation Standard Proposed |
| --- | --- | --- | --- | --- |
| Server Leaf | 16.01 ms | 3.91 ms | 0.07 | 0.06 |
| Border Leaf | 65.53 ms | 48.24 ms | 0.21 | 0.52 |

Numan et al. [36] also conducted research on conventional networks and SDN where they focused on the comparison of latency and jitter. The test carried out to obtain the latency value in this study was to calculate the round-trip delay in several scenarios. Meanwhile, in our research, measurements are based on reports generated from cacti tools. These are used daily to monitor throughput and latency in SDN and NFV network environments, so that the results produced are more accurate because the data is taken in real time.

*4.3. Manageability Evaluation*

In this test, four scenarios were used to evaluate the manageability of the SDN-NFV combination by calculating the duration of the test. The tests carried out were the integration between the SDN controller and the VM controller, adding an integrated policy for VM instances through the SDN controller and creating an endpoint group (EPG) on the SDN controller and port-group on the distributed virtual switch (DVS).

(1)    SDN and VM controller integration

The first test is to integrate the SDN controller with the VM controller, which is the beginning to combine SDN and NFV, where the two environments can be connected to each other. The initial requirements of this test are that the first VM controller has been formed, and then the interconnection between the server and the leaf switch server has also been established. After these two requirements are satisfied, integration can be initiated. To integrate the SDN controller with the VM controller, we simply need to do so in the SDN controller dashboard. Here we will create a special domain to accommodate the VM controller to be integrated by completing some basic VM controller information to be input on the dashboard. This is performed only once in one SDN controller sub-menu.

Based on the process flow described in Tables 7 and 8, we created a table of the estimated time required to complete each process, so that we could determine the estimated

total time required for each scenario. In Table 7, we can see that the estimated time needed to complete the integration scenario between the SDN controller and the VM controller is 3 min, including checking on the VM controller side. Whereas, if only SDN is used, the integration process becomes more complicated and takes longer, as shown in Table 8. This integration is also useful in the future, which will certainly make it easier for users when going to the process to set up the two environments.

**Table 7.** New VM controller integration measurement time via SDN controller.

| No | Process Name | Device | Measure Time (Minutes) |
|----|--------------|--------|------------------------|
| 1 | Access the Virtual Networking menu on the SDN controller to integrate the VM controller via the VMM-Domain sub-menu | SDN controller | 2 |
| 2 | Checking on the VM controller whether the integration folder has appeared | VM controller | 1 |

**Table 8.** Measuring time of new VM controller integration with SDN controller manually.

| No | Process Name | Device | Measure Time (Minutes) |
|----|--------------|--------|------------------------|
| 1 | Access the Virtual Networking menu on the SDN controller to create a VM domain controller manually via the VMM-Domain sub-menu | SDN controller | 5 |
| 2 | Access the Networking menu on the VM controller to create a DVS that will bind to the newly created VMM-Domain | VM controller | 5 |
| 3 | Access the Networking menu on the VM controller to create a port-group according to the DVS that has been created | VM controller | 3 |
| 4 | Access the Networking menu on the VM controller to create a port-group according to the DVS that has been created | SDN controller | 3 |

Table 9 compares the estimated time in minutes required to complete the scenario of adding a new instance with the increase in the number of instances. As the number of separate SDN and NFV environment instances increases, it takes more time than integrating SDN and NFV.

**Table 9.** VM controller integration measurement time comparison.

| Instance Number | SDN-NFV | SDN |
|-----------------|---------|-----|
| 1 | 3 | 16 |
| 2 | 6 | 32 |
| 4 | 12 | 64 |
| 6 | 18 | 96 |
| 8 | 24 | 128 |
| 10 | 30 | 160 |

(2)  Added integrated policies for VM instances

The second test adds an integrated policy for VM instances, which is very important because it is performed quite often when the virtualization environment is already onboard. Another aspect that is no less important than this test is to maintain policy consistency between SDN and NFV caused by human error. The policy that can be made from the SDN controller for the VM controller is an attachable access entity that contains information

about the physical interface between SDN and NFV, such as the discovery protocol interface, maximum transmission unit (MTU) and link aggregation. The interface policy regulates more towards layer 2 connections on the VM such as the spanning tree protocol (STP), port security, monitoring (Syslog) and Dot1x for authentication, if needed. All of these policies will be created entirely in the SDN controller, so that in the VM controller, we only need to check whether the policy we have created has been formed and register the host that will be bound to the policy created. This policy can be implemented in the attachable access entity profile, vSwitch policies, and leaf access port policy submenus.

In Tables 10 and 11, we can see that the process of adding policies in the SDN-NFV combination environment is sufficient to be done centrally on the SDN controller dashboard, whereas the process of adding policies to separate SDN and NFV networks takes more time because it requires manual configuration on the two different dashboards. Of course, apart from being time-consuming, a configuration that is not centralized is at risk of misconfiguration, which can also make deployment time longer because troubleshooting must be performed in both environments.

**Table 10.** Integrated policy increment measurement time for VM via SDN controller.

| No | Process Name | Device | Measure Time (Minutes) |
|---|---|---|---|
| 1 | Access the Access Policies menu on the SDN controller to create an Attachable Entity via the AEP Profile sub-menu | SDN controller | 5 |
| 2 | Access the Virtual Networking menu on the SDN controller to create a vSwitch Policy via the VMM-Domain sub-menu | SDN controller | 3 |
| 3 | Access the Access Policies menu *Policies* on SDN *controller* for create *Leaf Access Port Policy* via sub-menu *Leaf Access Port* | SDN controller | 5 |
| 4 | Access the Access Policies menu on the SDN controller to create a Leaf Access Port Policy via the Leaf Access Port sub-menu | VM controller | 1 |
| 5 | Access the Networking menu on the VM controller to add hosts to bind with the newly created policy | VM controller | 3 |

**Table 11.** Measuring time of policy increments with non-integrated SDN controller and VM controller.

| No | Process Name | Device | Measure Time (Minutes) |
|---|---|---|---|
| 1 | Access the Access Policies menu on the SDN controller to create an Attachable Entity via the AEP Profile sub-menu | SDN controller | 5 |
| 2 | Access the Networking menu on the VM controller for vSwitch configuration | VM controller | 5 |
| 3 | Access the Networking menu on the VM controller to create a Distributed Virtual Switch | VM controller | 5 |
| 4 | Access the Networking menu on the VM controller to map the network adapter to the host | VM controller | 15 |
| 5 | Access the Access Policies menu on the SDN controller to create a Leaf Access Port Policy via the Leaf Access Port sub-menu | SDN controller | 5 |
| 6 | Checking the VM controller whether the interface configuration is appropriate | VM controller | 1 |
| 7 | Access the Networking menu on the VM controller to add hosts to bind with the newly created policy | VM controller | 3 |

Table 12 compares the estimated time in minutes needed to complete the scenario of adding an integrated policy with an increase in the number of policies. As the number of

policies added increases, separate SDN and NFV environments require more time than integrated SDN and NFV environments in policy making.

**Table 12.** Comparison of measurement times for adding integrated policies.

| Instance Number | SDN-NFV | SDN |
|:---:|:---:|:---:|
| 1 | 17 | 39 |
| 2 | 34 | 78 |
| 4 | 68 | 156 |
| 6 | 102 | 234 |
| 8 | 136 | 312 |
| 10 | 170 | 390 |

(3)    Added Endpoint Group (EPG) and port-group

The next test is to add an endpoint group (EPG) to the SDN controller and port group. This test is important considering that this is one of the activities that administrators often perform when setting up the data center environment. With the rapid addition of applications to the company, it also requires speed of deployment on the infrastructure side. In this test, an endpoint group (EPG) is added, accompanied by the creation of a centralized port group on the SDN controller. The endpoint group (EPG) is an entity on SDN that contains information on server infrastructure or the like connected to SDN devices, such as the VLAN used and the port connected to SDN. To associate EPG with DVS, we can do this through the SDN controller dashboard in the application EPG sub-menu.

Based on Tables 13 and 14, we can see that the process of adding EPG and port-groups in the combined SDN-NFV environment is quite simple and can be done centrally on the SDN controller dashboard, while on separate SDN and NFV networks it takes more time because we need to perform manual configuration on two different dashboards. The time difference between the two tests was 3 min compared to 12 min. Of course, apart from being time-consuming, a configuration that is not centralized is very at risk of misconfiguration, which can also make deployment time longer because troubleshooting must be performed in both environments.

**Table 13.** EPG association measurement time with port-group.

| No | Process Name | Device | Measure Time (Minutes) |
|:---:|:---|:---:|:---:|
| 1 | Access the Tenants menu on the SDN controller to associate an EPG with a port-group via the Application EPG sub-menu | SDN controller | 2 |
| 2 | Checking on the VM controller whether the VLAN on the port-group matches the VLAN EPG | VM controller | 1 |

**Table 14.** Manually adding EPG and port-group measurement times.

| No | Process Name | Device | Measure Time (Minutes) |
|:---:|:---|:---:|:---:|
| 1 | Access the Tenants menu on the SDN controller to configure VLANs on the EPG sub-menu Application EPG | SDN controller | 5 |
| 2 | Access the Networking menu on the VM controller to configure VLANs on the port-group | VM controller | 3 |
| 3 | Access the Networking menu on the VM controller to bind port-groups that have been set to VLANs with DVS | VM controller | 3 |
| 4 | Checking on the VM controller whether the VLAN on the port-group matches the VLAN EPG | VM controller | 1 |

Table 15 provides a comparison of the estimated time in minutes required to complete the scenario of adding EPG and port groups with an increased number of instances. As the number of instances added increases, separate SDN and NFV environments take more time than integrated SDN and NFV environments to create instances.

**Table 15.** Comparison of measurement times of adding EPG and port-group.

| Instance Number | SDN-NFV | SDN |
|---|---|---|
| 1 | 3 | 12 |
| 2 | 6 | 24 |
| 4 | 12 | 48 |
| 6 | 18 | 72 |
| 8 | 24 | 96 |
| 10 | 30 | 120 |

If we combine the three scenarios in one graph, as shown in Figure 11, we can see that the time required for the combined SDN-NFV network to complete the task is faster than that for the separate SDN and NFV networks, as shown in Figure 18. We can understand this because on a combined SDN-NFV network, all configurations are performed centrally so as to minimize the need to configure or check individually for each device, which increase the deployment time and risks configuration errors and policy inconsistencies. In the research conducted by Awais et al. [37], SDN is a technology that is intended to simplify network management. They also said that it enables complex network configuration and easy controlling through a programmable and flexible interface. It opens a new horizon for network application development and innovation. Currently, the development of SDN integration is very rapid with various endpoints or workloads. One of the most influential is virtualization; this point is very important in implementing SDN deployments because currently there are many technology developments that lead to it, so that tight integration between SDN and virtualization platforms is needed. Therefore, in our research, we are very concerned about the integration and segmentation policy in the SDN environment and virtualization, specifically NFV.

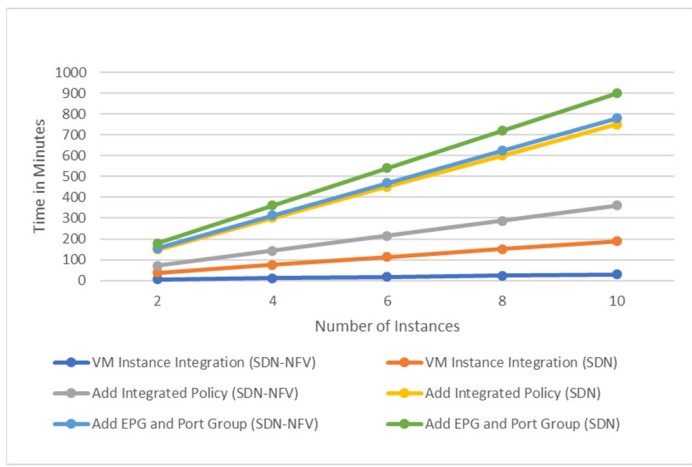

**Figure 18.** Manageability evaluation chart.

## 5. Conclusions

Speed and agility are indispensable in the application of 5G technology. The combination of SDN and NFV utilizes the concept of underlay and overlay in data forwarding, where the separation between the control plane and data plane is also performed. In this study, we proposed an integrated SDN-NFV architecture to simplify network management activities. We also modify the networking system at the computing level underlying NFV devices by replacing the default virtual switch with the Data Plane Development Kit

(DPDK) and Single Root I/O Virtualization (SR-IOV). After that, we evaluate SDN and SDN-NFV performance and manageability with the following results: performance increase of 190–200 Mbps for the server leaf and 0.8–1.6 Gbps for the border leaf. Meanwhile, the latency decreased by 12 ms for the server leaf and by 17 ms for the border leaf. For the manageability test with SDN and NFV integration, there were savings of 13 min for scenario 1, 22 min for scenario 2 and 9 min for scenario 3. This significantly reduces the time required to manage network devices by a factor of four. The results of this test indicate that the combination and modification of the architecture in this study can significantly improve data center network performance. This increase resulted from the acceleration of packet processing, starting from computing to capacity expansion at the data center networking device level. In terms of manageability, it also experienced efficiency in terms of operational activity time. This time efficiency is the result of the integration of the SDN and NFV Controller. Therefore, this research is in accordance with the intended technical goals. For future work, researchers can conduct research to discuss security in the integration of SDN NFV and its surrounding systems.

**Author Contributions:** Conceptualization, N.S. and N.A.P.; methodology, N.S.; software, N.A.P.; validation, N.A.P.; formal analysis, N.S.; investigation, N.A.P.; resources, N.S.; data curation, N.A.P.; writing—original draft preparation, N.S. and N.A.P.; writing—review and editing, N.A.P.; visualization, N.A.P.; supervision, N.S.; project administration, N.A.P.; funding acquisition, N.S. All authors have read and agreed to the published version of the manuscript.

**Funding:** This research publication is fully supported by Bina Nusantara University.

**Conflicts of Interest:** The authors declare no conflict of interest.

## Nomenclature

| | |
|---|---|
| Gbps | Unit of measurement data in a given amount of time, gigabits per second |
| Mbps | Unit of measurement data in a given amount of time, megabits per second |
| ms | Time, millisecond |

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
