# Peer review of "Integrated SDN-NFV 5G Network Performance and Management-Complexity Evaluation"

_futureinternet, doi:10.3390/fi14120378_

Round 1

Reviewer 1 Report

This paper targets a very interesting topic of SDN and NFV from practical and implementation point of view. The paper suggested some modifications also at the computing level of VNF devices.

The authors comprehensively compared the performance of separated SDN-NFV and integrated SDN-NFV in terms of throughput and latency.

However, the paper seems like a master practical project  and not a research work. It could fit for a demo session and conference papers I feel.

In the related work, a lot of text was used for explaining NFV and SDN which are well known technologies. This explanation is useful maybe in the project report. But not for MDPI journal papers.

In the related work section, several old NFV-SDN integration papers were also discussed. Based on this related work review, we cannot really evaluate how the idea of this paper is innovative.

Looking quickly for related works, I could suggest the authors look at the following papers or similar ones and improve the related work section in still their work is innovative.

https://onlinelibrary.wiley.com/doi/chapter-epub/10.1002/9781119857921.ch8

https://www.sciencedirect.com/science/article/abs/pii/S1389128622003395

In addition, there is a need to review the paper from English point of view.

 Some observations to address :

In the abstract, review next sentence please ('In this study, we propose ..' , then next lines, 'We also proposed a modification' )

In the abstract, review next sentence in line 17 : 'We also proposed a modification to the ...'

Probably, ‘of’ is missing between  modification and the networking

In the abstract , 13 min --> 13 minutes ...

Line 222 the “W” should be capitalized?

The formatting of table 1 is chunked up maybe it will be better to add an empty row to separate the text of different rows

Lines 460-462, Figure6 and Figure 7 à Figure 16 and Figure 17?

 Line 626 ‘more time. perform manual’ -- > more time to perform manual

Reviewer 2 Report

·         As reported in abstract, the authors tested various scenarios but what would be the outcome? No objective stated, no conclusive remarks?

·         Some sentences are not understandable like line.95 …collaboration?? Should re word

·         The authors should add more references in ‘intro’ section

·         Nomenclature should be added

·         Line.76 put ref for several techniques

·         The heading of section.2 Related work need to be change

·         All the sub-headings of section.2 should be merged and make it concise

·         Explain Spine-leaf architecture in 2-3 lines

·         Replacing SDN with open flow is obvious? Fig.2 shows what?

·         Again section.3 should be brief. Remove irrelevant stuff

·         Table.2 depicts what?

·         Fig.5 not clear

·         The propose SDN with NFV is new? Or it is modified?

·         Performance evaluation section, authors discussed many venues but it is better to make it in one table and all figures should define in one way

·         Conclusion section need to add all results of research

·         Overall the manuscript is quite lengthy with penalty of irrelevant details

Round 2

Reviewer 2 Report

should accept